# Interaction and influence of a flow field and particleboard particles in an airflow forming machine with a coupled Euler-DPM model

**Jian Zhang**, **Qing Chen***, **Minghong Shi**, **Hongping Zhou**, **Linyun Xu**

College of Mechanical and Electronic Engineering, Nanjing Forestry University, Nanjing, People's Republic of China

* qchen@njfu.edu.cn

**Data Availability Statement:** All relevant data are within the paper and its Supporting information files.

## Abstract

Particleboards are widely used in the artificial board market, which can be constructed from a variety of raw materials and require small amounts of energy to be produced. In the particleboard production process, forming machines play an important role as the key equipment for achieving continuous production. In recent years, airflow forming machines have received increasing attention in particleboard production lines because of their strong separation ability and low price. However, the internal flow field is complex and difficult to control, which affects the surface quality and strength of the particleboard. The most pressing technical difficulty is controlling the flow field characteristics of the airflow paver. At present, the research on this subject is conducted primarily through repeated experiments, which entail long research periods and high processing costs. To reduce human and financial costs, in this study, Computational Fluid Dynamics (CFD) is employed to investigate the flow field and the gas-solid two-phase flow field coupled with particle movement of an airflow forming machine. The accuracy of the calculation model is verified by comparing characteristic point velocities obtained from experimental analysis and a simulation. The simulation results show that in practical production, the frequency of a negative pressure fan should be greater than 27 Hz. It is necessary to set the shoulder properly, and the slab smoothness can be improved by moving the shoulder back on the premise of meeting the strength requirements of the box. The distance between the shoulders of the box body should be less than 2570 mm, and particles with uniform diameter should be added to the paving box to reduce the turbulence effect, improve the quality of particle forming and provide actual particleboard production with a solid theoretical foundation.

## Introduction

Particleboards are a kind of engineered wood, produced by chipping and grounding tree's logs in order to obtain the wood particles [1]. Particleboard possesses the advantages of sound absorption, heat insulation, high surface flatness and environmental protection performance. Therefore, particleboard is widely used in the furniture manufacturing, car manufacturing and

**Funding:** Qing Chen is supported by the National Natural Science Foundation of China (No. 51906111). He plays a role in study design, decision to publish and preparation of the manuscript.

**Competing interests:** The authors have declared that no competing interests exist.

construction industries. Particleboard has a varied source of raw materials [2, 3], low energy consumption and wide usage, which causes the demand for particleboard to increase annually. The increasing demand for particleboard at home and abroad has promoted the development of the wood-based panel industry, which has made the particleboard industry a growing business worldwide [4].

Recently, most of the research has focused on particleboard materials and the optimization of wood performance [5–10]. Moslemi et al. found that cellulose nanofibers can be extracted from rice straw and mixed with urea-formaldehyde adhesive for the production of medium density fiberboard (MDF) to optimize the rupture modulus, elastic modulus and internal bond strength of the MDF [11]. The sandpaper types, sanding time and density of rice straw particleboard have a clear effect on the surface roughness of the particleboard [12]. In addition, machine learning can be applied to predict the relevant properties of wood fiber boards for improved quality control in real time [13]. Numerous studies have indicated that the performance of particleboards is positively related to paving technology, therefore, research on particleboard paving equipment is important.

Mechanical pavement was used in the early particleboard production lines. The distribution of the throwing needles in the circumferential direction of the traditional mechanical pavement rollers was discontinuous, and the cylindrical throwing needles caused part of the shavings to be thrown at a certain angle in the tangential direction of the pavement rollers, resulting in unsatisfactory separation and pavement performance. With the development of science and technology, to meet the demand of market competitiveness and particleboard quality requirements, airflow forming machines and mechanical airflow mixing pavers have emerged. An airflow forming machine is based on the principle of gravity separation. When the adhesive shavings enter the pavement system, under the action of the air flow, the horizontal speed of the coarse shavings is small, the settlement is fast and the blowing distance is short, while the horizontal speed of the fine shavings is large, the settlement is slow and the blowing distance is long. Therefore, coarse and fine shavings are spread at different positions of the conveying belt to form the particleboard with a gradually changing structure [14]. Airflow pavement has low sensitivity to shaving shape, low price, and strong sorting ability for particleboards and can form slabs with good surface fineness and gradual section structures, which are widely used in particleboard production lines.

The technical difficulty of airflow forming machines is to control the characteristics of the airflow field. The characteristics have clear effects on the surface roughness, surface fineness and section gradient of the particleboard. However, little or no efforts have been made to study the airflow field characteristics of particleboard paver at present. To improve the pavement quality of particleboard, it is urgently important to study the characteristics of the airflow field. In previous studies, the flow field characteristics of the airflow forming machine were mainly obtained through experiments. The field test has a long research period and high processing cost. The application of Computational Fluid Dynamics (CFD) enables the development of airflow paver models, which provides a new method to study the characteristics of the airflow field.

The CFD method can analyze and display the phenomena that occur in the flow field, predict the performance in a short period of time and achieve the best design result by changing various parameters with low cost and high computational efficiency. Recently, the CFD method has been widely used in the wood and forestry field [15–17]. The drying process of wood particles is researched by coupling Discrete Element Method (DEM) and CFD models [18, 19], and the combustion characteristics of the packed bed in the wood chip combustor can be calculated and studied by CFD based on the Euler-Euler method [20, 21]. However, according to the literature retrieval, the CFD method has not been applied to analyze the

airflow field characteristics of airflow forming machines. The establishment of CFD simulation model of an airflow forming machine is a good alternative for or may complement field trials.

To analyze the airflow field characteristics and the gas-solid two-phase flow field of coupled particles in an airflow forming machine, a simulation model coupling with the Euler-DPM model was established in this study. The accuracy of the numerical model is verified by comparing the experimental data with the simulation data. The numerical simulation is performed in two phases: the airflow phase, which is carried out by solving the Navier-Stokes equations, and the particle transport phase, which is solved by tracking dispersed particles through the calculated airflow field. By adjusting various parameters, the airflow field characteristics under different conditions are obtained. The optimization analysis of the airflow forming machine is performed to improve the quality of particleboard pavement.

## Materials and methods

### Principle of particleboard airflow forming machine

The principle of a particleboard airflow forming machine is to pave sizing shavings by pneumatic separation. The shavings scattered in the airflow field of the paving machine are accelerated and further separated. The coarse shavings with larger diameters fall closer to the air inlet, while the fine shavings with smaller diameters fall farther away, forming slabs with gradually changing structures on the transport conveyer belt.

This study takes a self-developed particleboard airflow forming machine as an example, and the working principle is shown in Fig 1. The sizing shavings enter the measuring bin through the conveyor. When the amount of shavings in the measuring bin reaches the rated loading, the positive pressure fan and negative pressure fan are started. When the airflow field in the paving box is stable, the motors for the scattering roller and diamond roller are started simultaneously. Once the pavement instruction is sent out, the measure motor and the motor of the dispersal roller in the measuring bin are immediately started to achieve the blanketing and paving of the shavings. The positive pressure air volume generated by the positive pressure fan passes through the uniform air volume device and enters the vertical damper and the horizontal damper. The shavings falling into the pavement box are blown over under the action of the positive pressure fan and move to the tail of the pavement box with the flow. Because of gravity, coarse shavings settle faster than fine shavings. A part of the coarser shavings falls on the diamond roller directly under their own weight and are reverse transported to the outside of the pavement box falling on the transport conveyor belt, which plays the role of secondary separation.

### Airflow field test system of airflow forming machine

The airflow field test system of the paver consists of six main parts: a positive pressure fan, pavement box, multipoint anemometer probe, multipoint anemometer probe adjusting device, multipoint anemometer data acquisition system, computer and a negative pressure fan. In this study, the equipment used in the airflow field test system is the KANOMAX MODEL 1560 multipoint anemometer developed by Kayo Corporation of Japan. The probe for wind speed collection is connected to the anemometer and then connected to the computer equipped with the software for wind speed testing through the interface. The type of multipoint anemometer probe used in this study is 0964–01, the measurement range is 0.1 ~ 50 m/s, and the test precision is ±0.75 m/s.

As shown in Fig 2, the length and height of the test section are 2250 mm and 1390 mm, respectively. Positions 1, 2 and 3 are selected as the height test positions, and the number of multipoint anemometer probes is seven. Due to the wall effect, the wind speed at the edge of

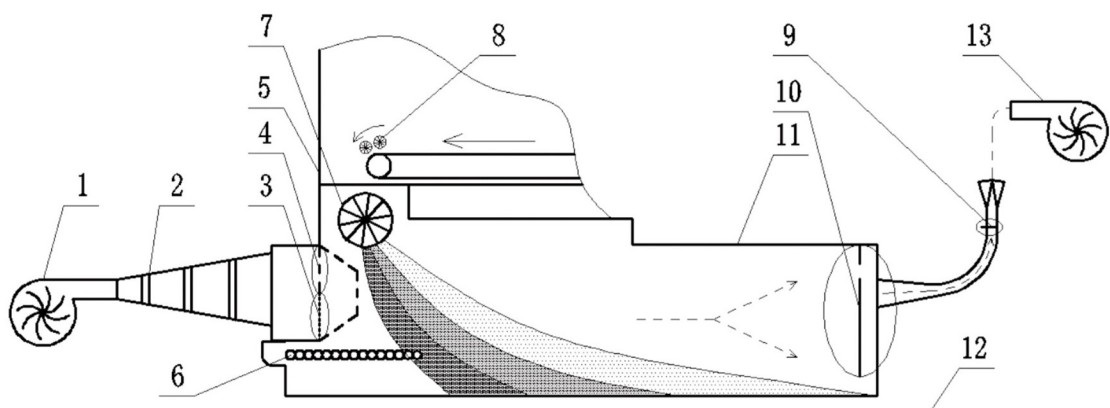

1. Positive pressure fan 2. Uniform airflow device 3. Horizontal flashboard 4. Vertical flashboard 5. Measuring bin 6. Diamond roller 7. Scattering roller 8. Dispersal roller 9. Tail flashboard 10. Tail airflow balance plate 11. Pavement box 12. Transport conveyor belt 13. Negative pressure fan

**Fig 1. Schematic diagram of the principle of the particleboard airflow forming machine.**

the airflow field is lower than that at the middle. To measure the changes in the edge flow field characteristics in more detail, the distance between the measuring points at the edge is set to be slightly smaller than that between the measuring points at the middle. The distance between the measuring points at the edge and at the middle are 300 mm and 400 mm, respectively. The probe can be adjusted in the height direction through the multipoint anemometer probe adjusting device. In the airflow field test experiment, a large amount of airflow field wind speed data is measured by changing the opening of the tail flashboard, horizontal flashboard and vertical flashboard to control the characteristics of the airflow field.

## Simulation and verification

In previous experiments, a multipoint anemometer was used to measure the airflow field velocity, and various parameters were adjusted to study the flow field characteristics of the airflow forming machine. In each experiment, it was found that many interrelated parameters were involved in the experiment, and the adjustment range of parameters was wide, therefore,

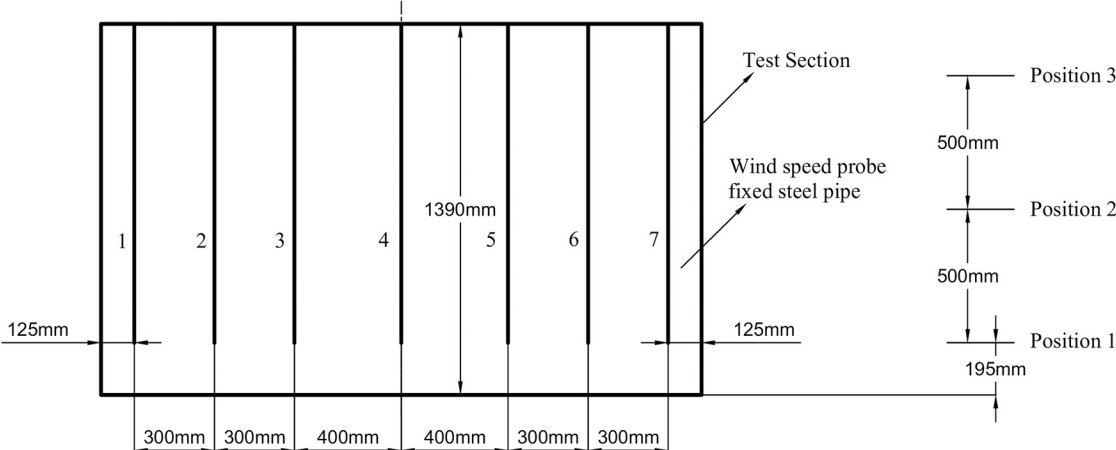

**Fig 2. Schematic diagram of the measuring point arrangement.**

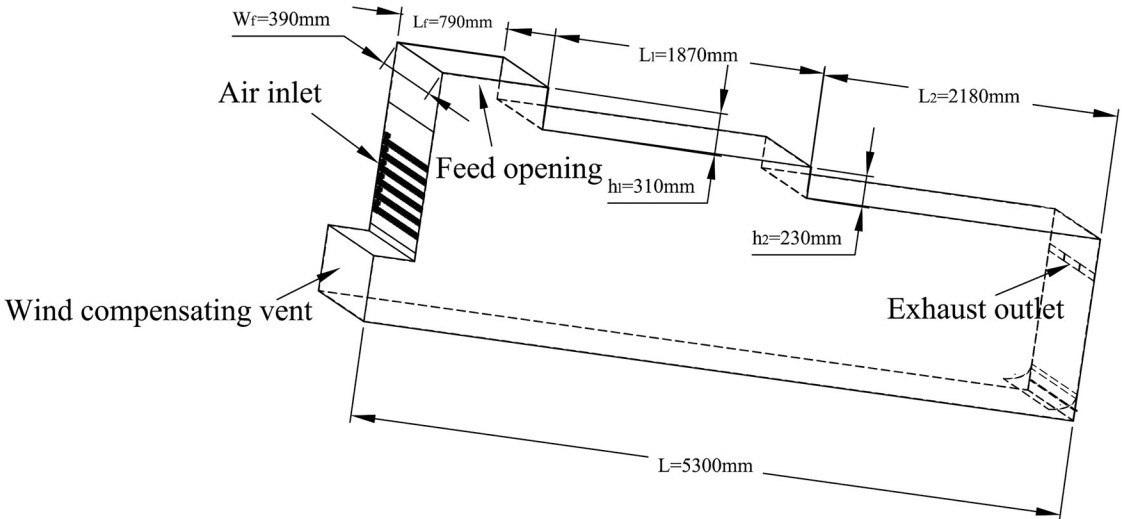

**Fig 3. Box model of airflow forming machine.**

the experimental period lasted too long, and the manpower and economic costs were large. In this study, the CFD method is used to analyze the airflow field of a particleboard airflow forming machine and the gas-solid two-phase flow field coupled with shavings particles.

## Airflow field simulation model and case settings

The airflow pavement cabinet model of this study is presented in Fig 3. The three-dimensional model of the box is composed of the air inlet, feed opening, wind compensating vent, exhaust outlet and wall surfaces. In the box model, the total length of the box is $L = 5300$ mm, the length of the first-step shoulder is $L_1 = 1870$ mm, the length of the second-step shoulder is $L_2 = 2180$ mm, the height of the first-step shoulder is $h_1 = 310$ mm, the height of the second-step shoulder is $h_2 = 230$ mm, the length of the feed opening is $L_f = 790$ mm, and the width of the feed opening is $W_f = 390$ mm. As shown in Fig 4, the air inlet (2650 mm×900 mm) is composed of 192 ventilation doors, each with a height of $H = 45$ mm and a width of $W = 20$ mm. The air inlet and exhaust outlet simulate the air intake of the positive pressure fan and the suction of the negative pressure fan, respectively. The wind compensating vent simulates the makeup air of the box. The setting of the velocity and turbulence intensity of each part is determined by the field conditions. The gridding of airflow forming machine is shown in Fig 5.

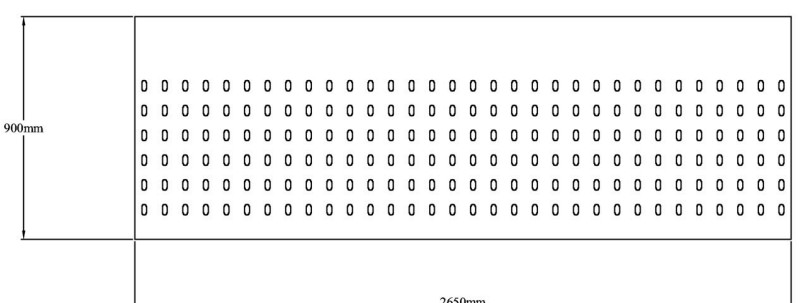
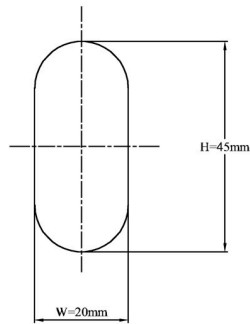

**Fig 4. Air inlet size.**

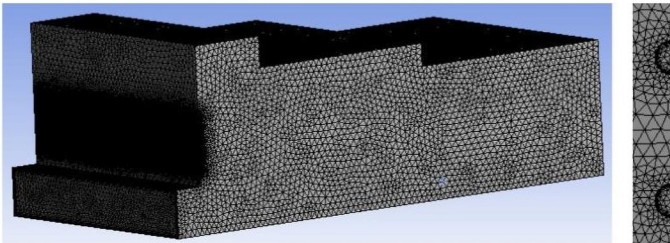

**Fig 5. Mesh generation and mesh refinement.**

Because the size of the air inlet and exhaust outlet are smaller than that of the whole box, the mesh of the air inlet and exhaust outlet is refined, which can ensure the quality of the mesh and improve the accuracy of the simulation.

For the flow field characteristics of the airflow forming machine studied in this paper, the air velocity of the flow field is less than 100 m/s, therefore, it can be regarded as an incompressible fluid. The continuity equation and momentum conservation equation can be written as

Continuity equation:

$$\nabla \vec{u} = 0$$

Momentum equation:

$$\nabla \cdot (\rho \vec{u} \vec{u}) = -\nabla p + \mu \nabla^2 u$$

In the formula, $\rho(kg/m^3)$ is the density of the air, $u\ (m/s)$ is the instantaneous velocity, $p(N/m^2)$ is the force of airflow on shavings, and $\mu(Pa \cdot s)$ is the aerodynamic viscosity coefficient.

In this study, Fluent 16.0 software was used to simulate the internal flow field of the airflow forming machine. The boundary conditions of the airflow forming machine model are set as follows: ① The air inlet is velocity-inlet, and the air inlet speed can be obtained from the frequency of the positive pressure fan; ② The wind compensating vent is pressure-outlet, and the pressure can be set to atmospheric pressure; ③ The exhaust outlet is velocity-inlet, and the exhaust outlet speed can be obtained from the frequency of the negative pressure fan; and ④ The wall condition is that the velocity on the solid surface satisfies a slip-free boundary condition. In the initial model of this study, the frequencies of the positive pressure fan and negative pressure fan are 20 Hz and 27 Hz, respectively, measuring that the wind speeds of the air inlet and the exhaust outlet are 10.4 m/s and 24.7 m/s, respectively.

The air diffusion in the airflow forming machine is slight, and the standard $k - \varepsilon$ model is suitable for slight diffusion, which is widely used in industrial flow field simulation, so the standard $k - \varepsilon$ model is used to calculate the three-dimensional flow field inside the airflow forming machine. The computational domain is discretized by means of a finite volume method [22]. The spatial discretization, including momentum and turbulent parameters, follows a second-order upwind scheme. In this approach, higher-order accuracy is achieved at cell faces through a Taylor series expansion of the cell-centered solution about the cell centroid. The flow field of the airflow forming machine is in a steady state, and the SIMPLE algorithm is widely used, which easily converges; therefore, the treatment of the pressure-velocity coupling is carried out by means of the SIMPLE algorithm [23]. The convergence criteria chosen for the simulation is to obtain a minimum normalized residual value of $10^{-4}$ for all the parameters.

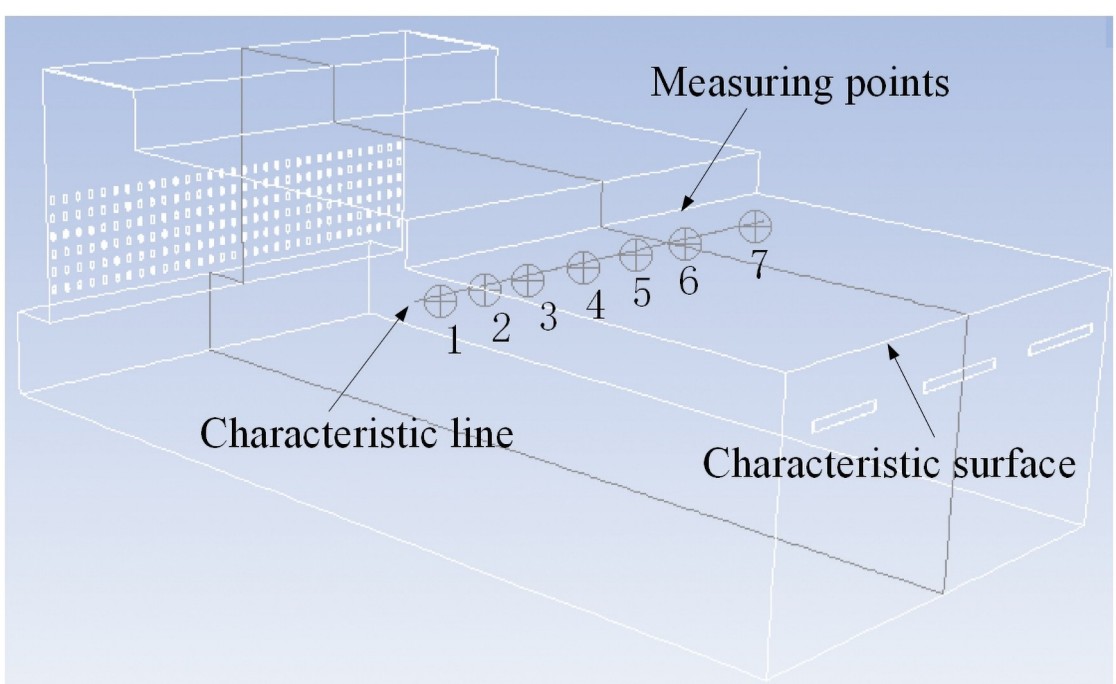

**Fig 6. Selection of measuring points, characteristic lines and characteristic surfaces.**

## Verification and simulation results

**Verification.**  Seven measuring points are selected on the characteristic line of the airflow forming machine (Fig 6) to compare the simulated and experimental values of the wind speed at the seven points, as shown in Table 1. The error is maximum for point 1; however, the <5% error is acceptable for the initial simulation analysis. Therefore, the accuracy of the selected numerical model can be verified, which can be used for the analysis of the flow field of an airflow forming machine.

**Original model simulation results.**  Fluent 16.0 is used to simulate the airflow field under the original model and initial boundary conditions. The characteristic surface is selected to analyze the flow field velocity vector and total pressure distribution of different models, as shown in Fig 6.

The 3D streamline plot of airflow in the paving box is illustrated in Fig 7. By combining the velocity vector (Fig 8) and the total pressure distribution of the extracted characteristic surface (Fig 9), it can be observed that the airflow direction in the forming machine is relatively stable,

**Table 1. Comparison of simulated and experimental values.**

| Measuring point number | Test value/ ($m \cdot s^{-1}$) | Numerical value/ ($m \cdot s^{-1}$) | Error/% |
|:---:|:---:|:---:|:---:|
| 1 | 2.05 | 2.10 | 2.44 |
| 2 | 2.38 | 2.41 | 1.26 |
| 3 | 2.56 | 2.59 | 1.17 |
| 4 | 2.47 | 2.48 | 0.40 |
| 5 | 2.64 | 2.64 | 0 |
| 6 | 2.60 | 2.55 | 1.92 |
| 7 | 2.20 | 2.19 | 0.45 |

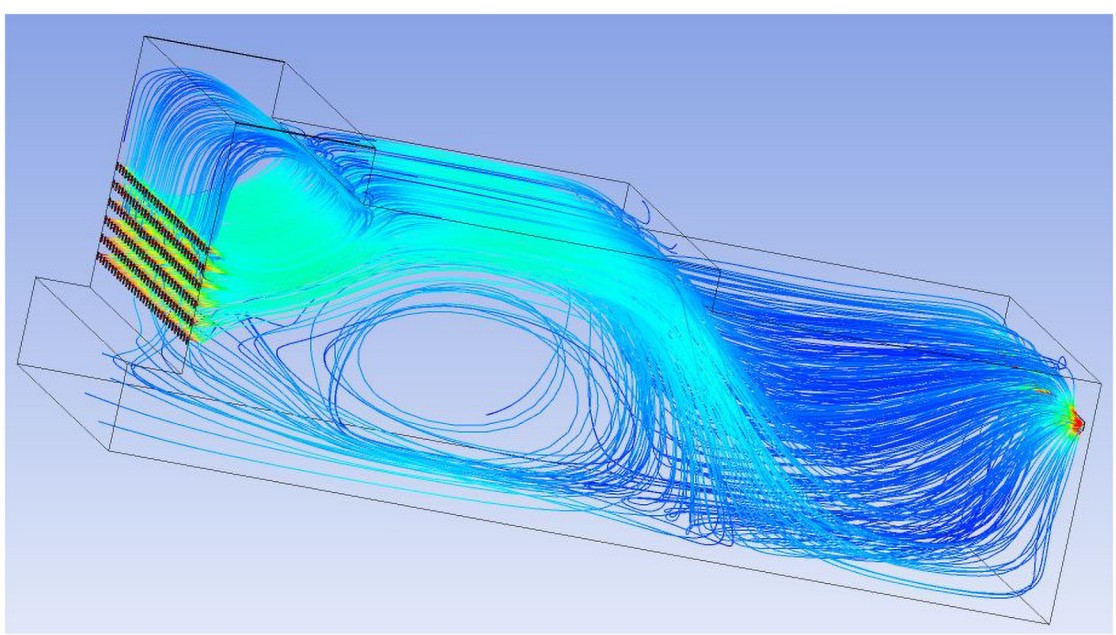

**Fig 7. Three-dimensional airflow field streamline diagram.**

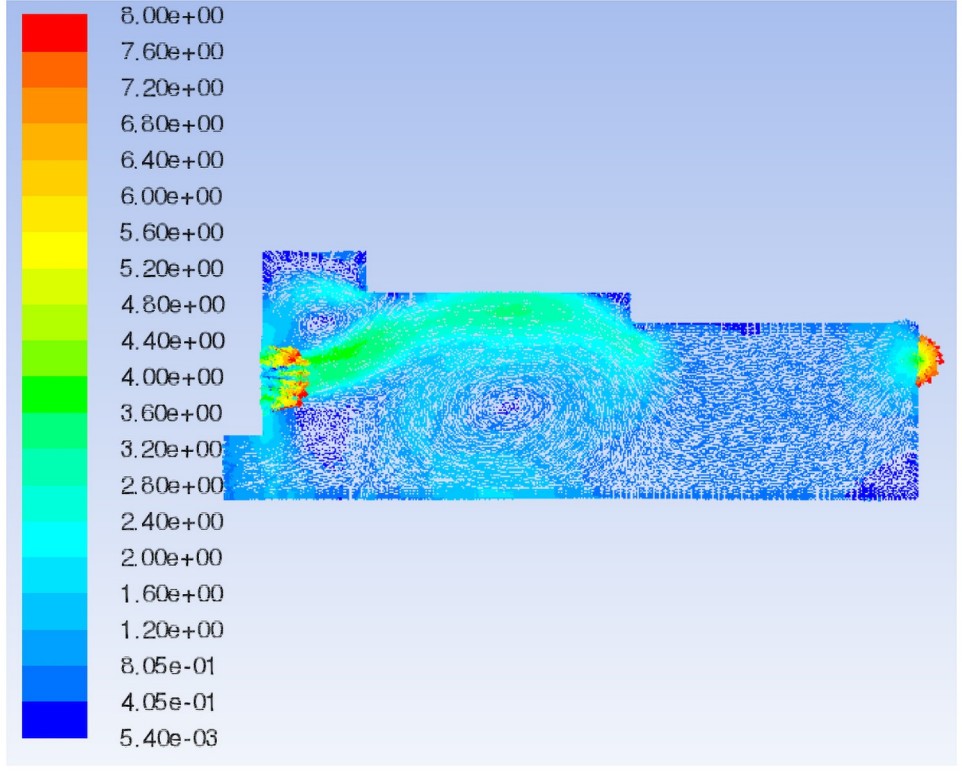

**Fig 8. Flow field velocity vector of the original model.**

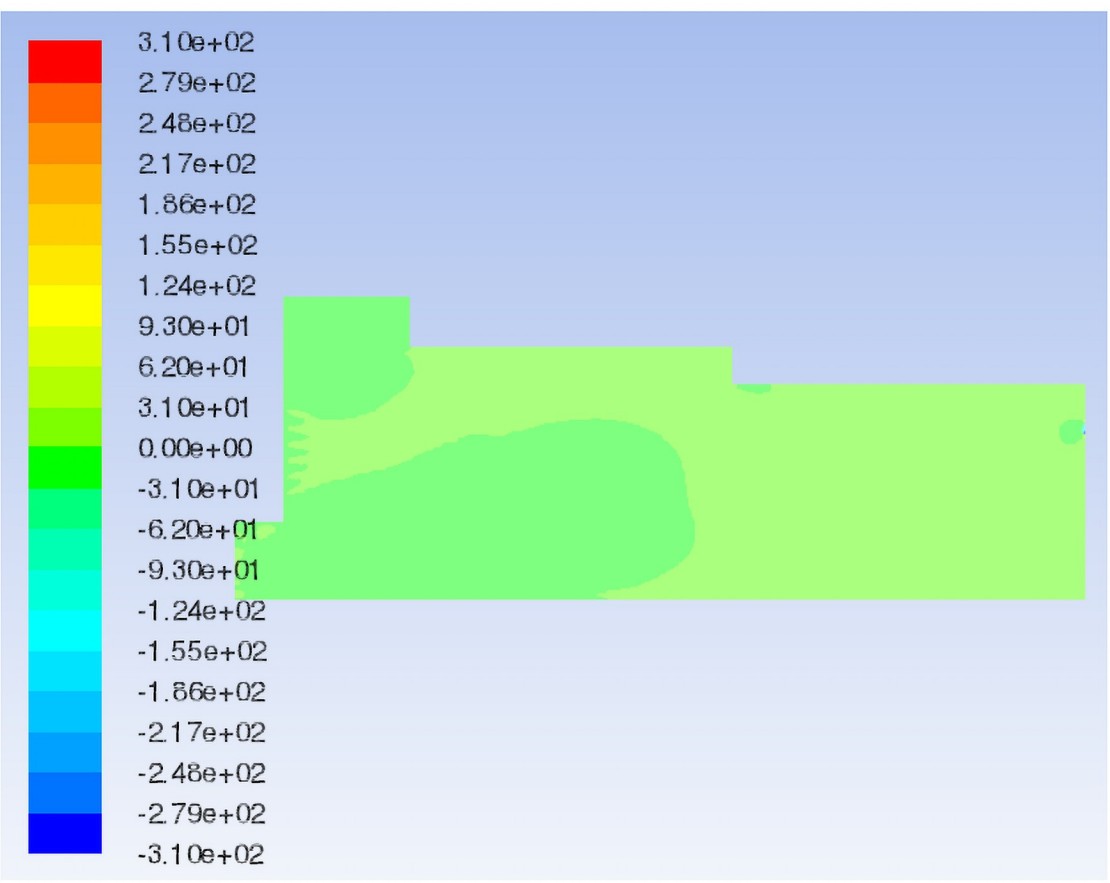

**Fig 9. Flow field total pressure distribution of the original model.**

and the air at the tail of the box is concentrated. There are eddies at the feed opening and the center of the box, which leads to turbulence and affects the quality of the slab pavement. Airflow at the sudden changed sections of the box is relatively concentrated, and the pressure is greater than that in the surroundings. At the same time, the pressure at the tail of the box is slightly larger than that at the front; that is, the pressure at the exhaust outlet is greater than that at the air inlet.

## Influence of negative pressure fan air volume on the airflow field

In this study, the air volume of the negative pressure fan is adjusted by altering its frequency. Based on the original model, the air volume of the negative pressure fan is increased and decreased to study the influence of the negative pressure fan on the air flow field. The frequency and air volume of the negative pressure fan are shown in Table 2.

**Table 2. Different negative pressure fan parameters and outlet speeds.**

| Model | Fan frequency/$Hz$ | Fan air volume/$m^3 \cdot h^{-1}$ | Outlet speed/$m \cdot s^{-1}$ |
|---|---|---|---|
| Original model | 27 | 12065.8 | 24.7 |
| Increasing air volume | 45.7 | 19543.7 | 40 |
| Decreasing air volume | 15.2 | 7328.8 | 15 |

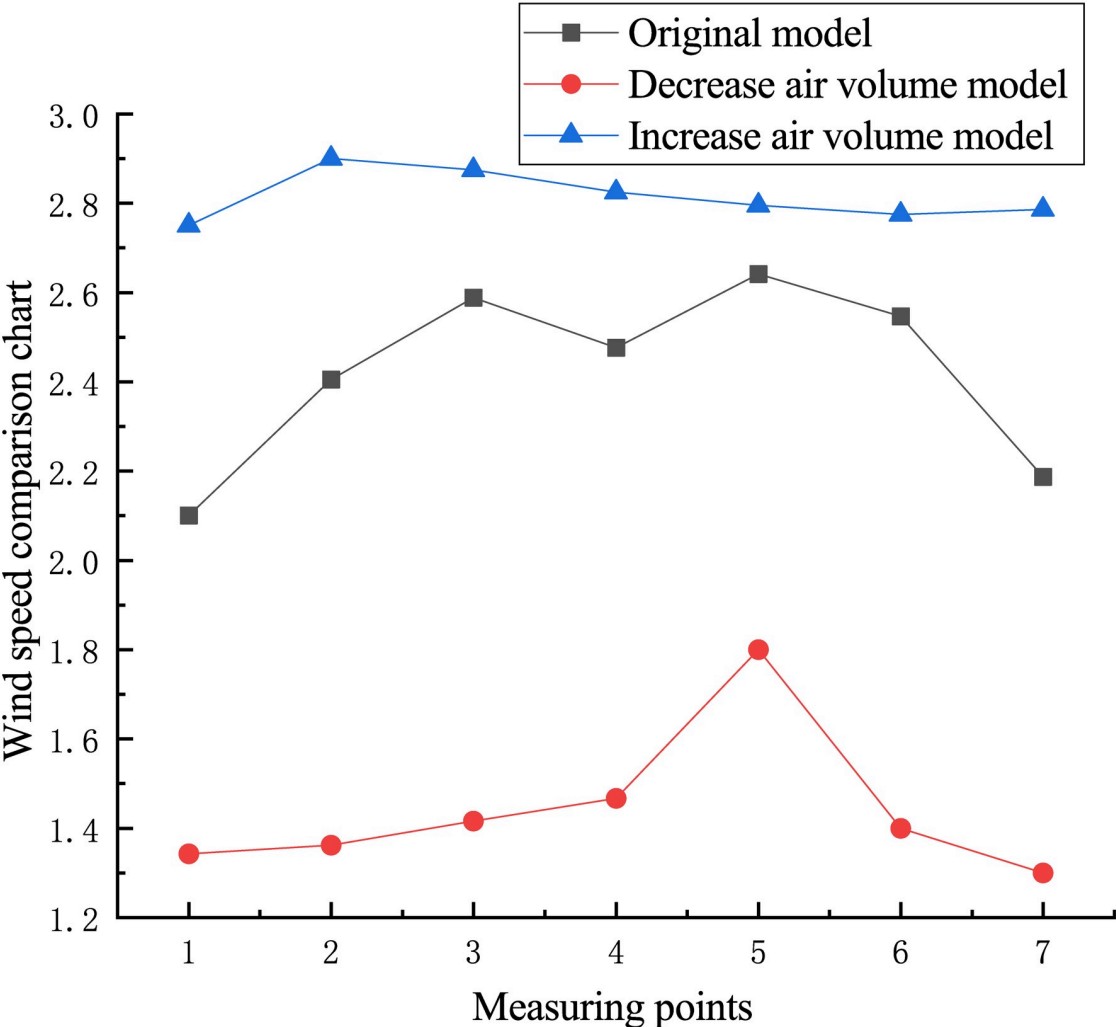

**Fig 10. Wind speed comparison of different models.**

In the models of different negative pressure fan air volumes, the speeds at the seven measuring points are compared by selecting seven points at the same position as the original model, as shown in Fig 10. The wind speed alteration at the measuring points is consistent with that of the negative pressure fan air volume. With the increase in the negative pressure fan air volume, the speed alterations of the seven measuring points are relatively stable, which improves the air flow stability. However, with the decrease in the negative pressure fan air volume, the speed of measuring point 5 alters abruptly due to the aggravation of turbulence, which reduces the air flow stability.

In the decreasing air volume model, the characteristic surface is selected at the same position as the original model to compare the velocity vector of the airflow field, as shown in Figs 8 and 11. When the air volume of the negative pressure fan is reduced, the air flow generated by the positive pressure fan is not quickly sucked out by the negative pressure fan due to the defects in the mechanical structure and the sudden changed sections of the box such that the pressure in the paving box increases. Part of the air flows out from the wind compensating vent at a higher speed, causing the material port to overflow. In addition, in the increasing air

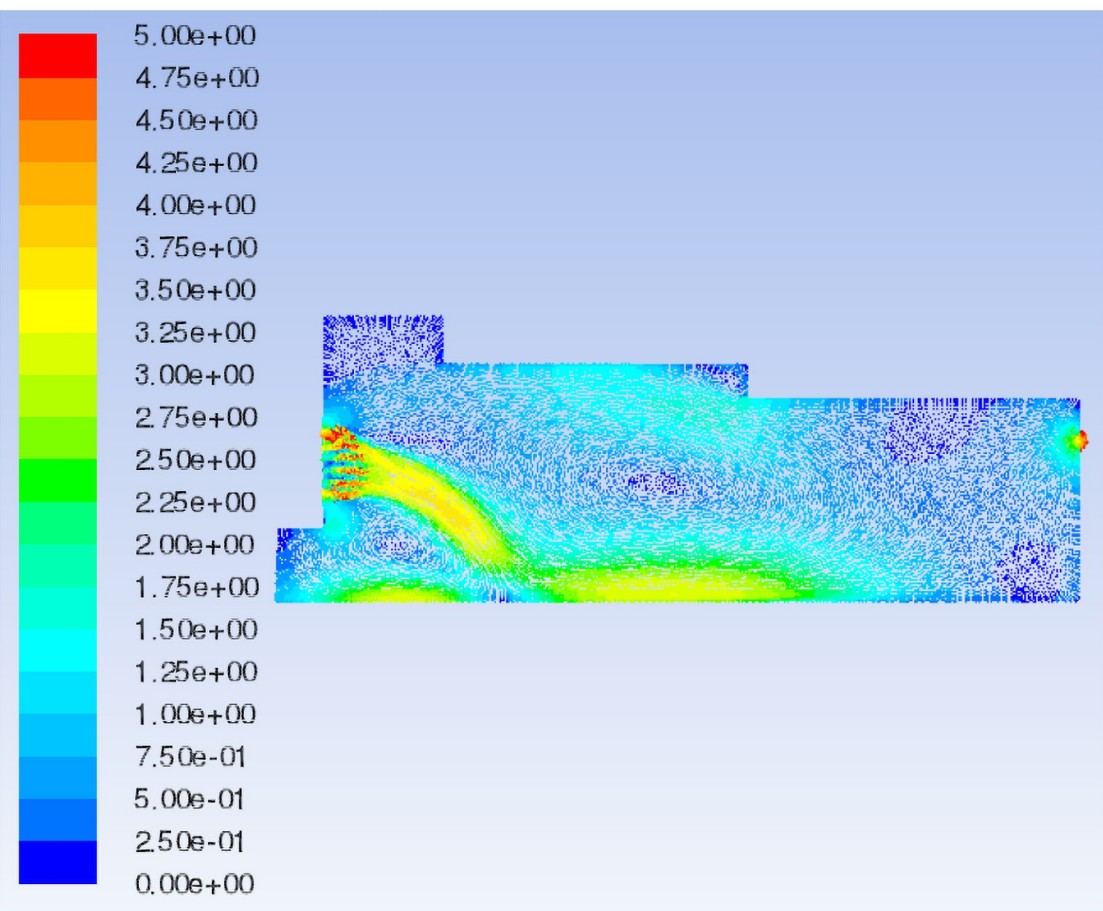

**Fig 11. Flow field velocity vector of the reduced air volume model.**

volume model, the same characteristic surface is selected to compare the velocity vector of the airflow field, as shown in Figs 8 and 12. When the air volume of the negative pressure fan is increased, the airflow direction in the paving box is more stable, forming a stable airflow direction channel. The turbulence caused by the airflow eddies at the feed opening and the middle of the box is also alleviated due to the increase in airflow velocity. However, at the second sudden changed section, a large amount of air hits the box. Comparing the flow field total pressure distribution before and after increasing the negative pressure fan air volume, as shown in Figs 9 and 13, it is clear that the pressure at the place where the airflow is concentrated from the first sudden changed section to the second reaches the maximum in the pavement box. At the same time, the pressure at the tail of the box gradually increases.

In the actual production process, to meet the strength requirements of the cabinet and avoid damage to the exhaust outlet, the air volume of the negative pressure fan cannot be increased indefinitely to improve the stability of the airflow. By changing the negative pressure fan frequency several times, it can be concluded that the negative pressure fan frequency should be at least 27 Hz. In other words, the minimum wind speed at the exhaust outlet is 24.7 m/s to sufficiently avoid the overflow at the wind compensating vent. The required specific frequency of the negative pressure fan should be determined according to the actual production situation.

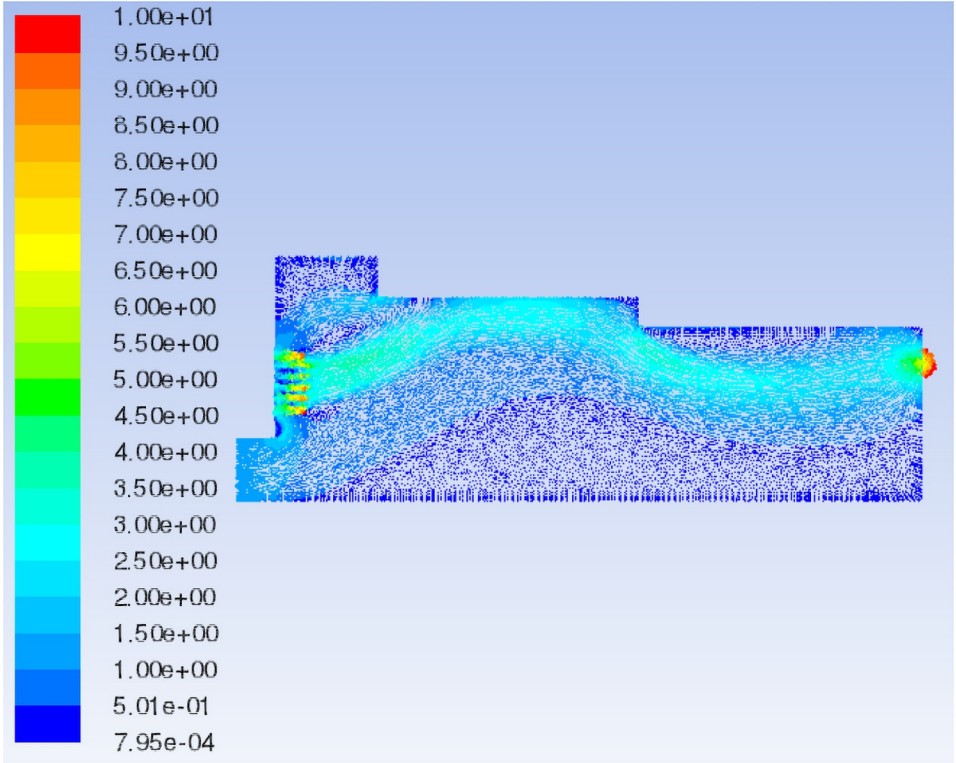

**Fig 12. Flow field velocity vector of the increased air volume model.**

### Influence of box geometry on airflow field characteristics

In this study, the original model is called Model 1. On the basis of Model 1, Models 2–5 with different geometric structures are established to study the influence of box geometry on the airflow field characteristics. The geometric parameters of the different models are shown in Table 3.

**Influence of box shoulder on the airflow field.**   In this section, the box shoulder-forward Model 2 and shoulder-backward Model 3 are selected to study the influence of the box shoulder on the airflow field.

The characteristic surface is selected at the same position as Model 1 to compare the velocity vector of the airflow field in Models 2 and 3, as shown in Figs 8, 14a and 14b. When the shoulder of the box is moved forward, the airflow eddies radius at the feed opening increases and the turbulence is aggravated such that the airflow stability in the box decreases and the airflow in the middle of the box swirls clearly. When the shoulder of the box is moved backward, eddies at the feed opening move to the right, and eddies shapes change from a circle to an ellipse, decreasing the radius of turbulence caused by the air eddies. The airflow eddies in the middle of the box almost disappear, and the air flows more steadily in the box with the shoulder moved backward.

Compare the flow field total pressure distribution of Models 1–3, as shown in Figs 9, 15a and 15b. When the box shoulder is moved forward, owing to the airflow eddies, the pressure in the box shows a downward trend as a whole. When the box shoulder is moved backward, the pressure in the airflow concentrated area and at the tail of the box increases. Therefore, the

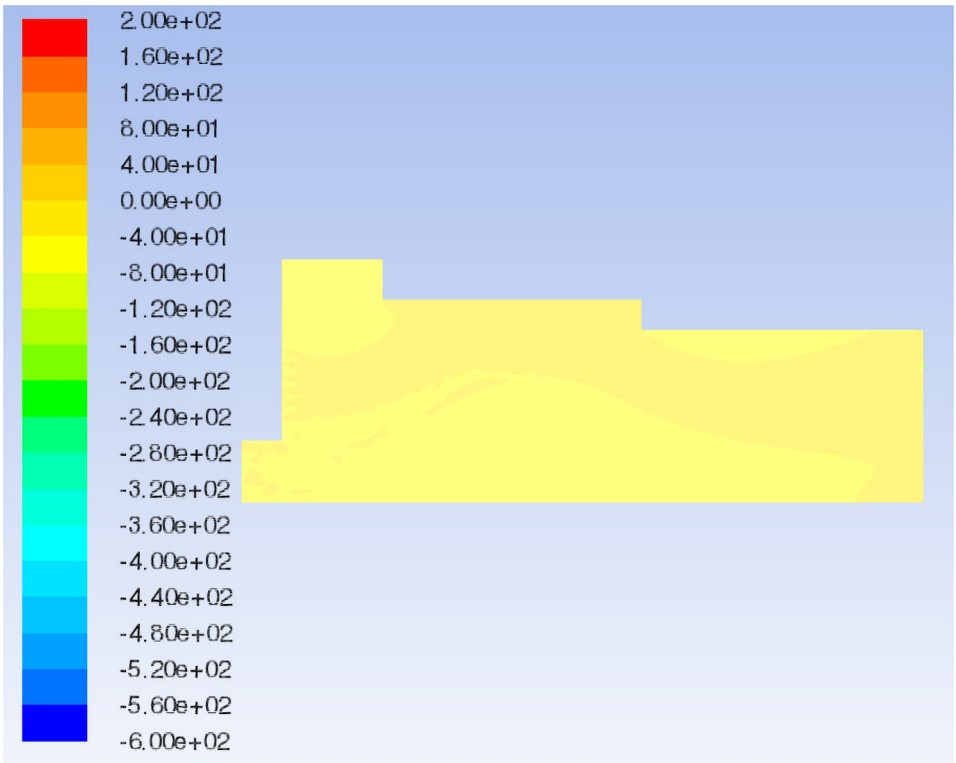

**Fig 13. Flow field total pressure distribution of the increased air volume model.**

back movement of the shoulder increases the pressure in the box, which is particularly pronounced at the tail of the box.

According to the airflow field simulation analysis of Models 2 and 3, it can be seen that the backward movement of the shoulder can improve the stability of the airflow to improve the paving quality of the particleboard. To obtain better airflow field characteristics, the distance of the shoulder is gradually increased. When the length of the shoulder is increased to 2570 mm, the airflow field characteristics in the box alter. The box shoulder backward movement Model 4 is established for analysis.

The flow field velocity vectors of Models 1, 3 and 4 are compared, as shown in Figs 8, 14b and 14c. With the increase in the backward movement distance of the box shoulder, eddies radius at the feed opening gradually decreases, and the turbulence is reduced by degrees, making the airflow direction centralized and stable. However, when the length of the shoulder increases to 2570 mm, the airflow eddies in the middle of the box reappear due to the gradual decrease in air velocity, affecting the quality of slab paving. The flow field total pressure

**Table 3. Geometric parameters of different models.**

| Model | $L_1$/(mm) | $h_1$/(mm) | $h_2$/(mm) | L/(mm) |
|---------|---------|---------|---------|---------|
| Model 1 | 1870 | 310 | 230 | 5300 |
| Model 2 | 1370 | 310 | 230 | 5300 |
| Model 3 | 2370 | 310 | 230 | 5300 |
| Model 4 | 2570 | 310 | 230 | 5300 |
| Model 5 | 1870 | 310 | 0 | 5300 |

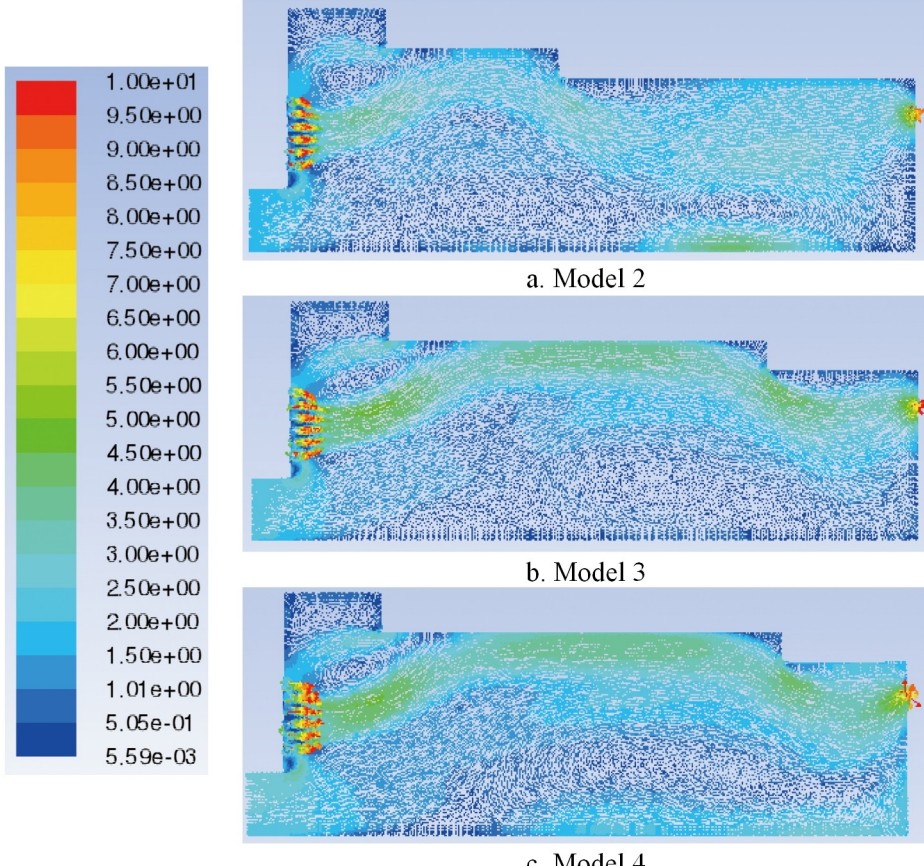

**Fig 14. Flow field velocity vector of different paver models.**

distribution of Models 1, 3 and 4, is compared in Figs 9, 15b and 15c. With the increase in the backward movement distance of the shoulder, the pressure at the airflow concentration and the tail of the box gradually increases. When the distance of the box shoulder increases to 2570 mm, the airflow eddies in the middle of the box decrease the pressure.

Based on the comparison and analysis of the above simulation, it can be seen that compared with the forward movement of the shoulder, the backward movement of the shoulder is beneficial to reduce the turbulence caused by the airflow eddies, improving the paving quality of the slab. However, the back movement of the shoulder causes a significant increase in the pressure at the tail of the box. In addition, when the shoulder is moved back too far, it leads to eddies in the middle of the box and reduces the quality of the slab paving. Therefore, the length of the shoulder should be less than 2570 mm.

**Influence of box shoulder removal on the airflow field.**   In this section, on the basis of Model 1, the shoulder of the airflow forming machine is removed. Model 5 is selected to study the influence of box shoulder removal on the airflow field characteristics.

The characteristic surface at the same position as Model 1 is selected to compare the velocity vector of the airflow field in Model 5, as shown in Figs 8 and 16. When the shoulder is removed from the box, the airflow eddies radius at the feed opening is reduced, and eddies in the middle of the box are also relieved. Most of the air flows from the air inlet to the exhaust outlet along the upper wall of the box, and the airflow is fairly uniform and stable because of the disappearance of the sudden changed section. Nevertheless, the airflow concentration at

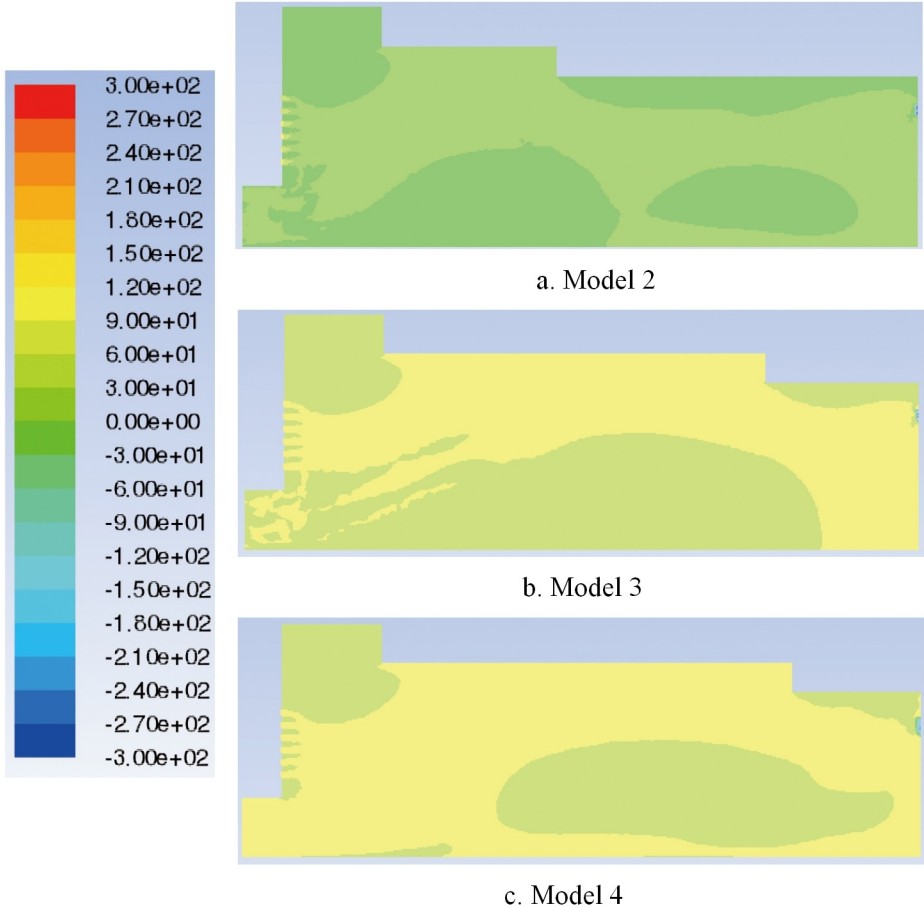

**Fig 15. Flow field total pressure distribution of different paver models.**

the air inlet and exhaust outlet is obvious, which means an increase in pressure. By comparing the flow field total pressure distribution of Models 1 and 5 (Figs 9 and 17), it can be seen that when the shoulder is removed from the box, the pressure in the box increases significantly, especially at the airflow concentration, which brings a large loading to the box and easily causes damage to the exhaust outlet.

In the actual design of the airflow forming machine, to decrease the total pressure in the box and save raw materials, it is necessary to set up shoulders to increase the contact area between the airflow and the wall surface. The shoulder can be moved back properly on the basis of Model 1, but the distance between the shoulders of the box should be less than 2570 mm to ensure better airflow field characteristics, thereby reducing the effect of turbulence and improving the quality of slab paving.

## Simulation analysis of particle deposition based on the Euler-DPM model

To verify that good airflow field characteristics contribute to improving the pavement quality of particleboard, in this study, the particle deposition of different airflow paving machines is simulated and compared based on the Euler-DPM model. The influence of shaving particle addition on the airflow field and various parameters affecting the particle deposition effect are

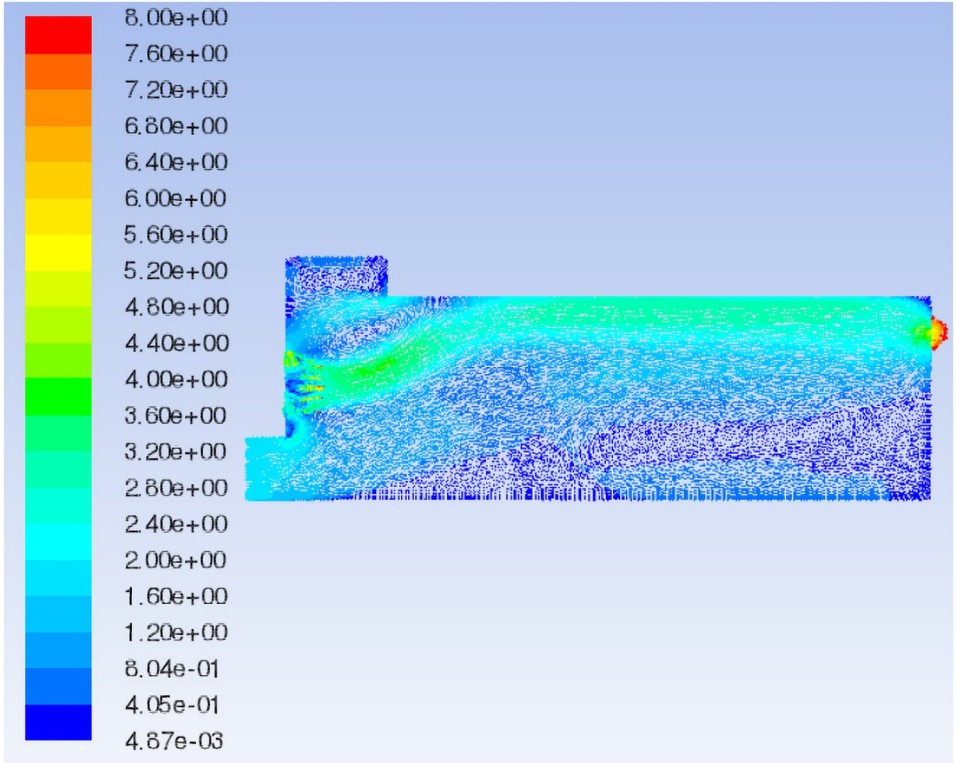

**Fig 16. Flow field velocity vector of Model 5.**

analyzed.

## DPM

A Discrete Phase Model (DPM) is utilized to study the transport process of shaving particles in the airflow forming machine. In this study, the Euler method is used to describe the airflow field and it is coupled with the Euler-DPM model to simulate the gas-solid two-phase flow field with shavings. The shaving particle trajectory is tracked by the unidirectional coupling Lagrangian method. The airflow field is calculated first, and then the force of the flow field on the particle is calculated. The equations governing the air flow for the internal field of the airflow paving machine are shown below:

Continuity equation:

$$\nabla \vec{u} = 0$$

Momentum equation:

$$\nabla \cdot (\rho \vec{u} \vec{u}) = -\nabla p + \mu \nabla^2 u + \vec{F}$$

The particles are injected at the feed opening. The trajectory of the particles in the flow domain is obtained by integrating the force balance on the particle. The force balance equating

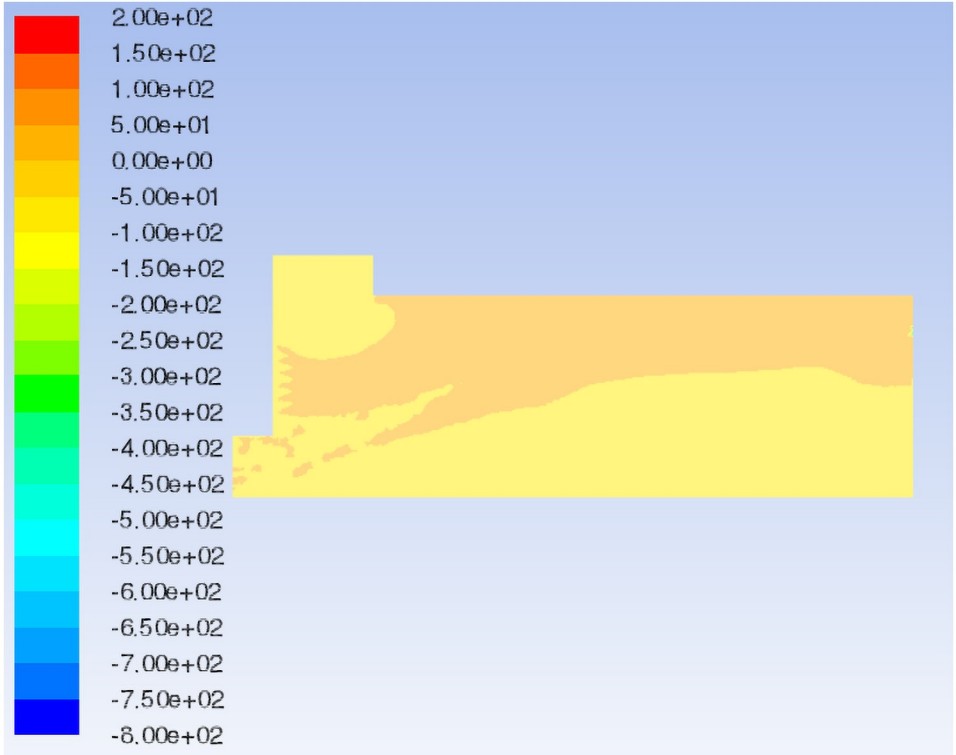

**Fig 17. Flow field total pressure distribution of Model 5.**

the inertia of the particle to the forces acting on the particle is given below:

$$\frac{du_p}{dt} = F_D(u - u_p) + \frac{g(\rho_p - \rho)}{\rho_p}$$

where $F_D(N)$ is the unit mass drag force generated by the coupled airflow field, $u(m/s)$ is the velocity of the airflow field, $u_p(m/s)$ is the particle velocity, $\rho(kg/m^3)$ is the airflow density, and $\rho_p(kg/m^3)$ is the density of the particles.

The momentum transfer from the fluid to the particles is obtained by computing the change in momentum of the particle as it passes through each control volume as given below:

$$\vec{F} = \sum \left( F_D\left(u - u_p\right) + \frac{g\left(\rho_p - \rho\right)}{\rho_p} \right) \dot{m}_P \Delta t$$

In this study, the DPM model boundary conditions are set as follows (Fig 18): The Discrete Phase BC Type of air inlet, wind compensating vent and exhaust outlet is set to escape, which means that the particles can pass through these three openings; the DPM boundary condition of the particle deposition surface at the bottom of the cabinet is set to trap, which means that the particles are captured on this bottom surface; the DPM boundary condition of other wall surface is set to reflect, which means that the particles rebound when they collide with these walls. Shaving particles are injected into the paving box from the feed opening, and the particles are set to spherical particles without an initial speed. In the actual production process, the quantity of shavings added is 3190.5 kg/h, so the mass flow is set to 0.88625 kg/s.

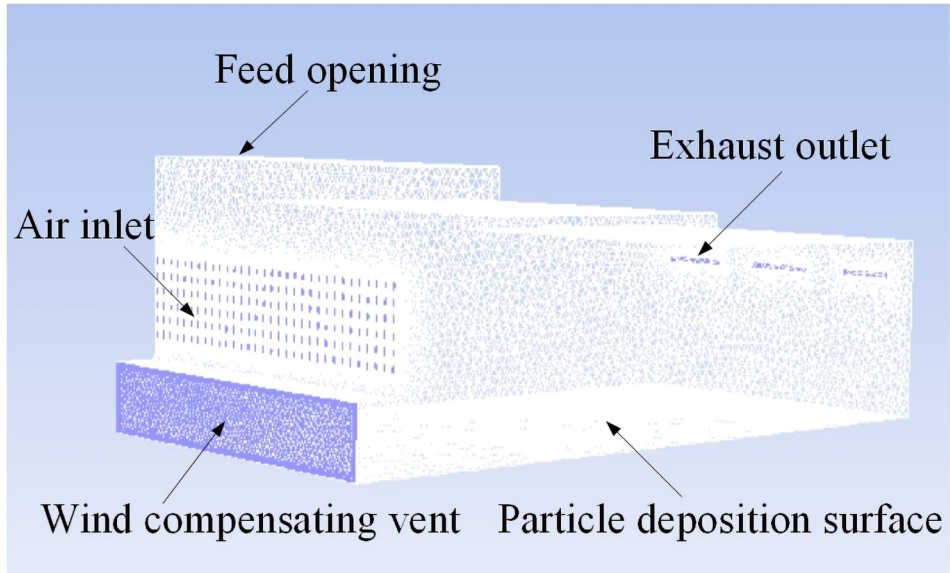

**Fig 18. DPM model settings.**

### Analysis of DPM simulation

**Influence of particle diameter distribution on deposition.** Shaving particles with a concentrated distribution and a normal distribution of particle diameters are injected into the airflow forming machine, and the particle diameters are in the range 0.0002 m-0.001 m. The distribution of the particle diameter and the particle trajectory of the corresponding DPM model are shown in Figs 19 and 20, respectively.

Comparing Fig 20a and 20b, it can be seen that when the diameter of the added particles is normally distributed, the fine particles of the same diameter deposit farther away from the bottom of the box, avoiding dense deposition. In addition, this approach reduces the mixing of

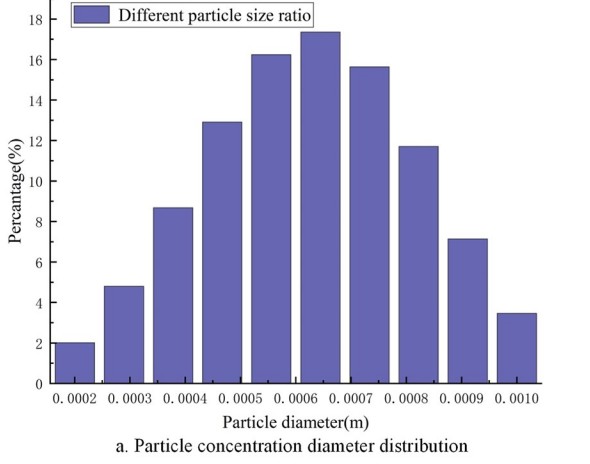

a. Particle concentration diameter distribution

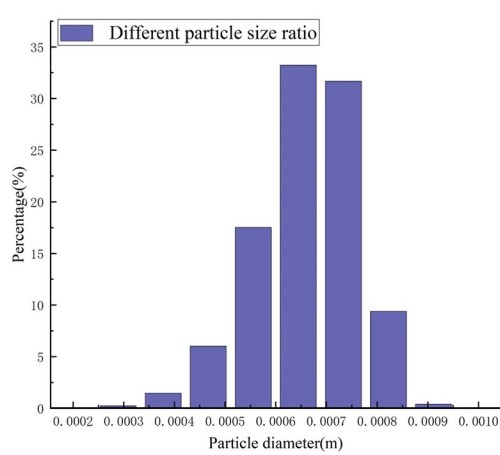

a. Particle concentration diameter distribution

**Fig 19. Particle diameter distribution.**

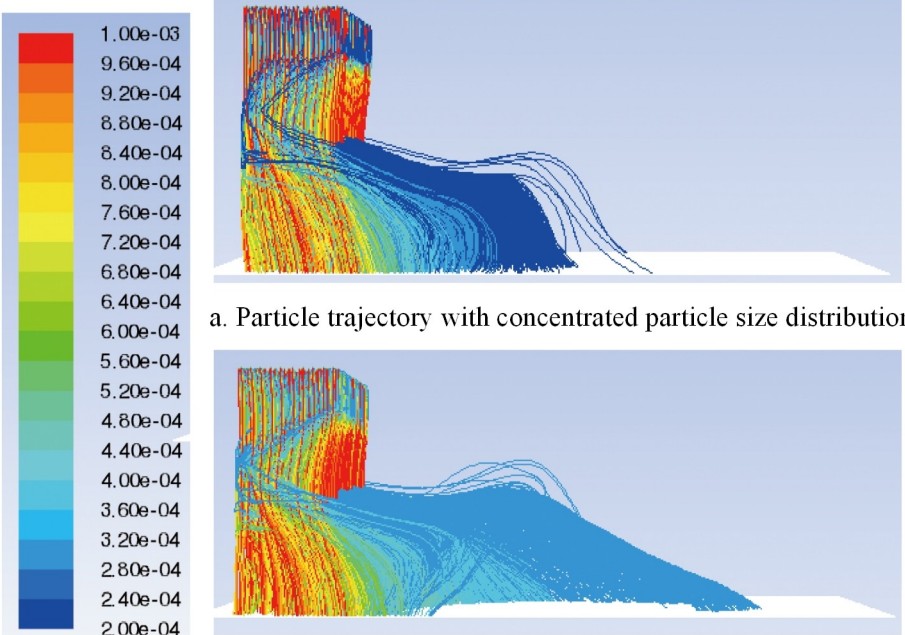

Fig 20. Particle trajectory of different particle size.

coarse particles of different particle sizes on the front deposition, thereby improving the uniformity of particle deposition on the bottom surface.

**Simulation of the airflow forming machine with shaving particles.** The characteristic lines are selected at the same position as the original model in the model with shaving particles, as shown in Fig 21 (Line-1 in the original model corresponds to the particle model Line-1*, and the other characteristic lines have the same correspondence relation).

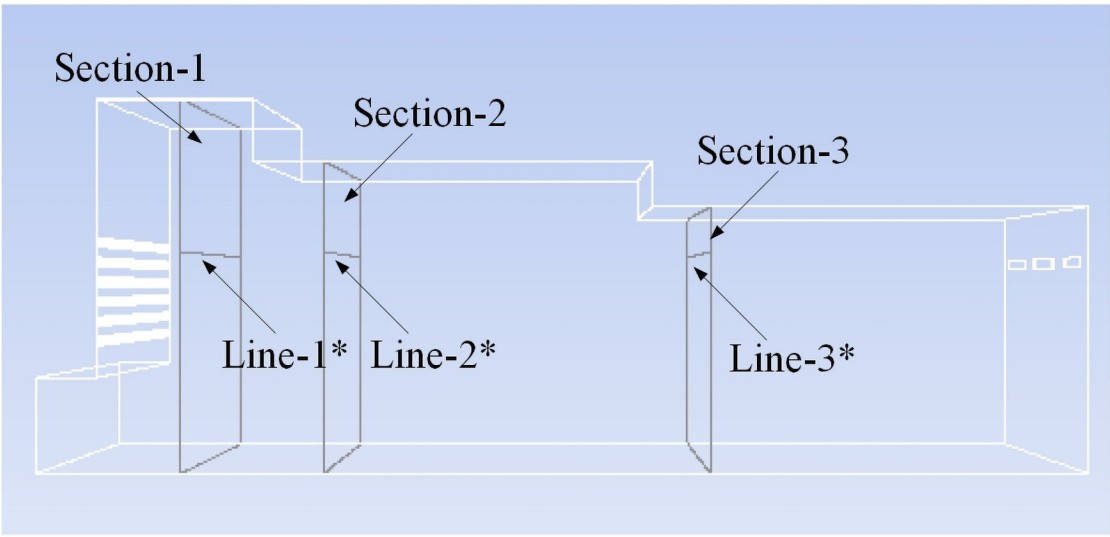

Fig 21. Selection of characteristic lines.

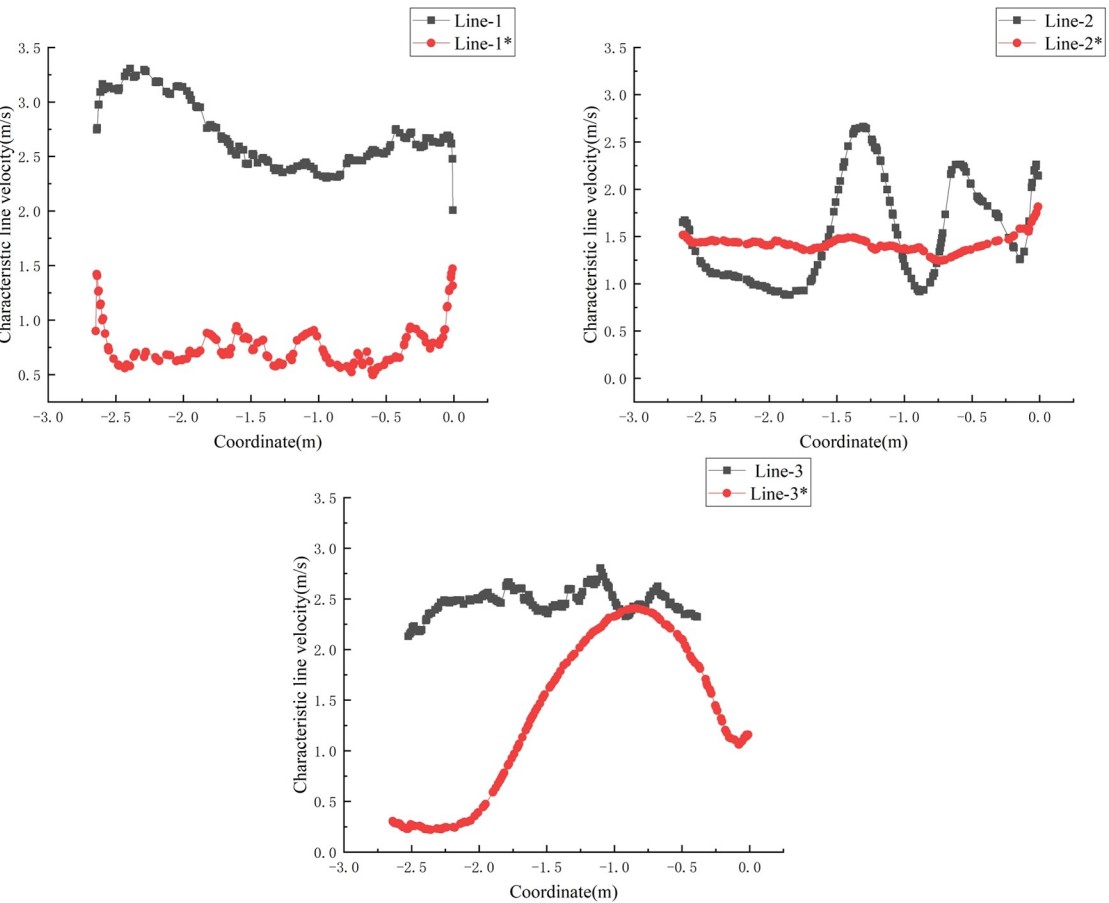

**Fig 22. Comparison of the characteristic line speed of the same section.**

Comparing the velocity variation of characteristic lines at different sections, as shown in Fig 22, it can be concluded that when shaving particles are added into the airflow forming machine, the characteristic line speed of each section is mostly lower than that of the original model. It is obvious from Figs 21 and 22 that Sections 2 and 3 are close to the sudden changed sections of the pavement box, which aggravates the turbulence of Sections 2 and 3. Therefore, the speed of the characteristic lines on Sections 2 and 3 changes obviously due to the addition of shaving particles, but the variation rule of the characteristic line speed in Section 1 is basically unchanged, and the range of velocity variation is stable. Therefore, it can be preliminarily concluded that when particles are added into the box, the effect on the airflow field at the tail of the box is greater than that on the front of the box.

In this study, the characteristic surface at the same position as the original model is selected in the particle model. When the particle diameter of the added shavings is in a normal distribution, the airflow field velocity vector and the total pressure distribution are shown in Figs 23a and 24a, respectively. Comparing the flow field velocity vector of the paving box before and after adding materials (Figs 8 and 23a), it can be seen that when particles are added to the paving box, the airflow eddies at the feed opening move to the right, and the eddies radius increases. The turbulence of the airflow at the sudden changed section is more serious than that of the original model, but the airflow at the tail of the box is more stable and uniform. Comparing the flow field total pressure distribution in Figs 9 and 24a, it can be concluded that

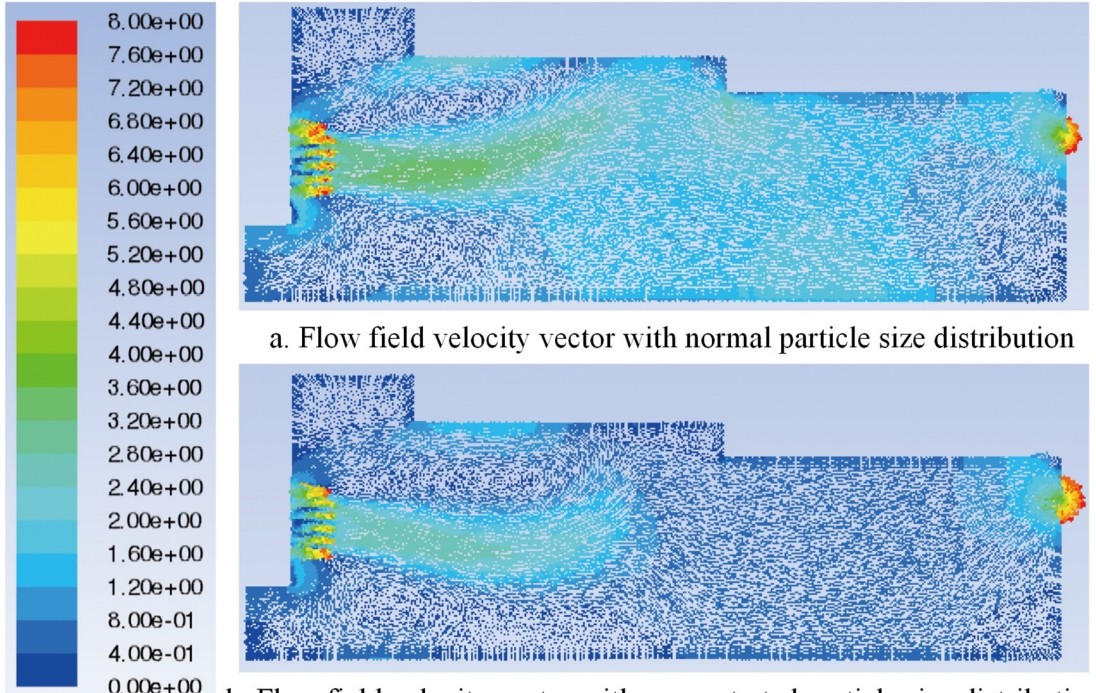

a. Flow field velocity vector with normal particle size distribution

b. Flow field velocity vector with concentrated particle size distribution

**Fig 23. Velocity vector of the airflow field with different particle sizes.**

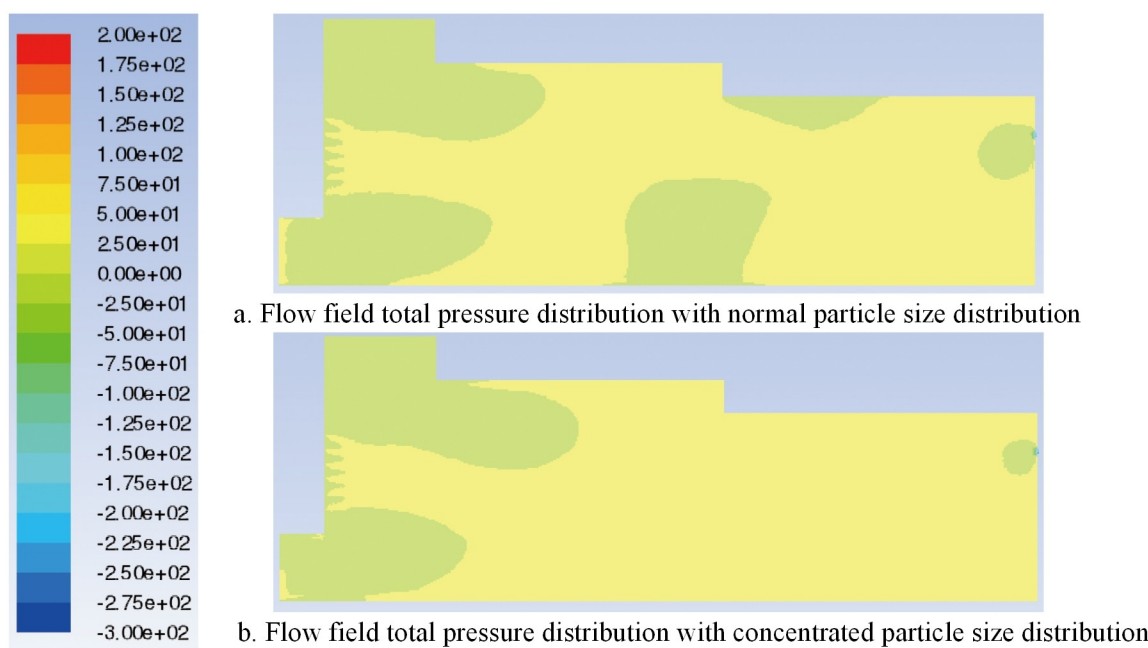

a. Flow field total pressure distribution with normal particle size distribution

b. Flow field total pressure distribution with concentrated particle size distribution

**Fig 24. Flow field total pressure distribution with different particle sizes.**

the whole pressure in the paving box increases, especially at the location where the airflow is concentrated.

Thus, when designing the box geometry structure, it is necessary to consider the increase in the pressure on the box caused by the addition of shaving particles.

When the particle diameter distribution of the added shavings is concentrated, the airflow field velocity vector and the total pressure distribution are shown in Figs 23b and 24b, respectively. Comparing the flow field velocity vector in Fig 23a and 23b, it can be seen that when the particle diameter distribution added to the paving box is concentrated, the airflow eddies at the feed opening move to the right again, and the eddies radius further increases. The turbulence of the airflow at the sudden changed section is more serious than that of the normal particle distribution model, resulting in a decrease in the stability of the airflow in the paving box. Comparing the flow field total pressure distribution in Fig 24a and 24b, it can be concluded that when the particle diameter distribution is concentrated, the pressure at the tail of the box increases more than that of the model with a normal particle diameter distribution, which may cause damage to the exhaust outlet.

In the actual production process, to improve the quality of the slab and reduce the loading of the box, it is necessary to add shavings with relatively homogeneous particle diameters.

**Influence of box geometry on particle deposition.** Based on the simulation analysis above, both the backward movement and the removal of the shoulder are beneficial to improve the airflow stationarity. Among all the shoulder backward movement models, Model 3 with a shoulder distance of 2370 mm is the relatively optimal model. Therefore, Model 3 is selected as the representative of the shoulder backward movement model, and the deposition of particles in the box is compared with Model 5 of shoulder removal.

When shaving particles with a normal distribution diameter are added into the airflow forming machine, the particle trajectories of Models 3 and 5 are compared, as shown in Fig 25a and 25b. When the shoulder is moved backward or removed, the deposition of particles on the bottom of the box is uniform, and there is little difference in deposition between Models 3 and 5. In this simulation, 6220 particles are traced in total. In Model 3, 6064 particles deposit on the bottom surface, and 156 particles escape through the exhaust outlet, while in Model 5, 5330 particles deposit on the bottom surface, and 890 particles escape through the exhaust outlet. Compared with Model 3, the number of particles escaping through the exhaust outlet significantly increases in Model 5, which leads to more waste of raw materials. Therefore, Model 3 is better, which is consistent with the above conclusion.

**Experimental results analysis.** Under the present processing conditions, the airflow forming machine corresponds to Model 1. The frequency of the negative pressure fan is adjusted while the geometric parameters remain unchanged. When the frequency of the negative pressure fan is greater than 27 Hz and less than 27 Hz, the field experiment pavement conditions are shown in Fig 26a and 26b, respectively.

Comparing Fig 26a and 26b, it can be concluded that when the frequency of the negative pressure fan is greater than 27 Hz, the airflow paver forms a slab with good surface fineness and a gradual section structure, improving the pavement quality. This is consistent with the simulation results, which verifies the accuracy of the CFD model in this study.

## Conclusions

In this study, a CFD model of a particleboard airflow forming machine is established to simulate the airflow field and the gas-solid two-phase flow field coupled with shaving particles. Generally, the results obtained with the simulation coincide with those obtained under field conditions. Consequently, the model can be considered an appropriate tool for optimizing the

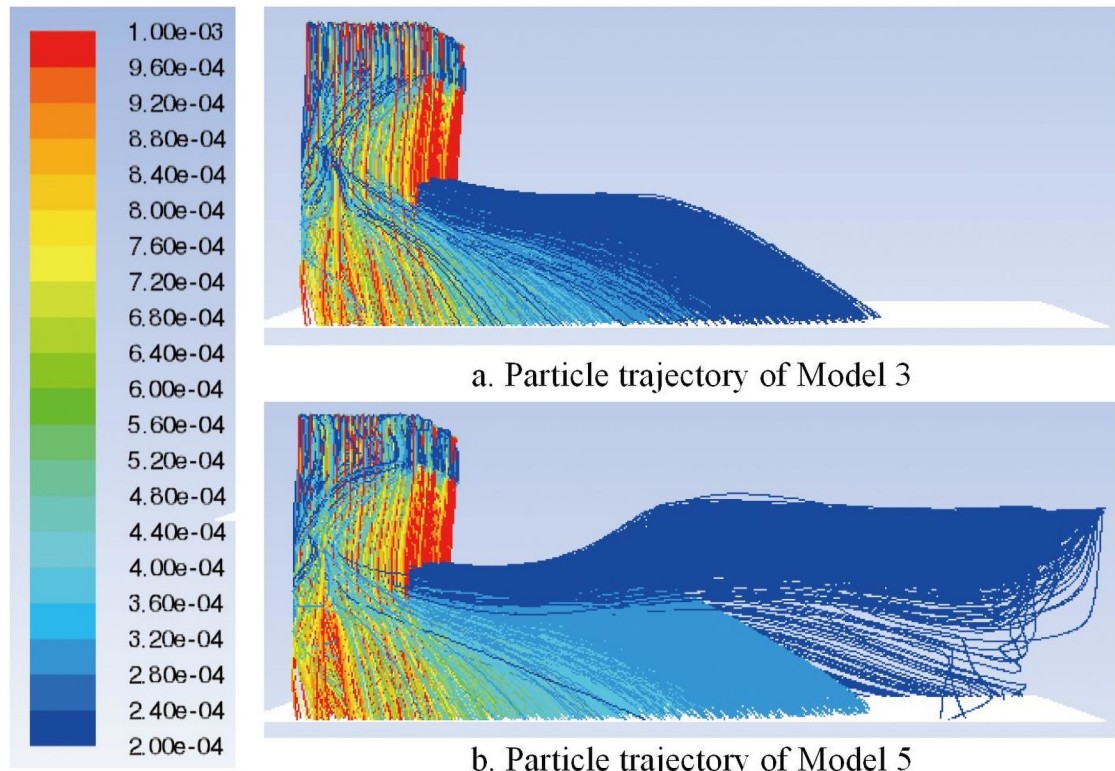

Fig 25. Particle trajectory of different box structures.

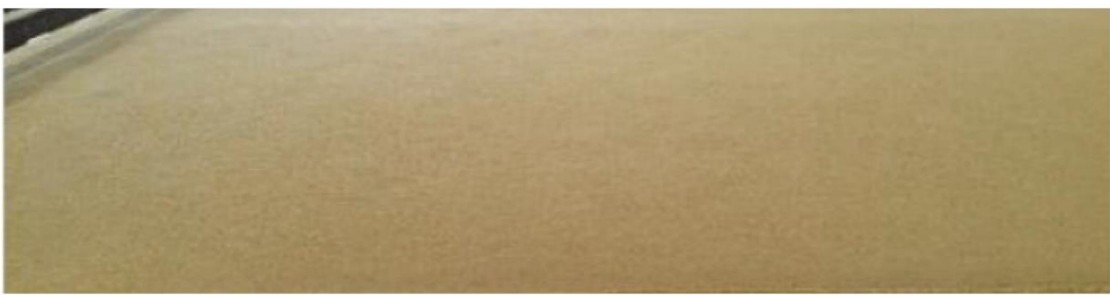

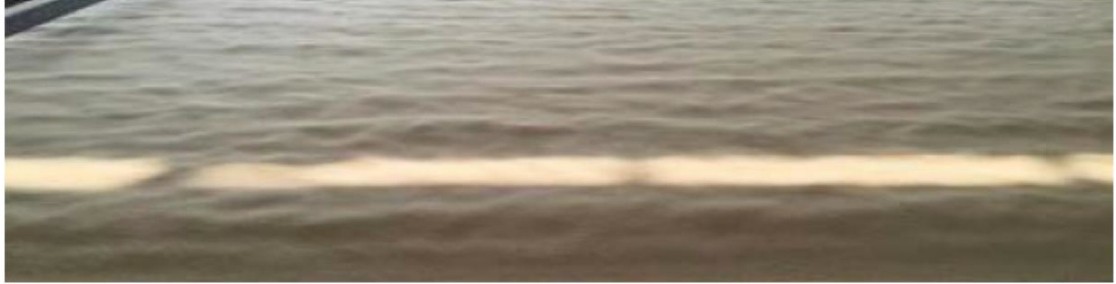

Fig 26. Pavement condition of different negative pressure fan frequencies.

structure of the paving box and improving the quality of slab paving. It may also serve to study different factors affecting the quality of slab paving, such as the air volume of the negative pressure fan, the box geometry and the particle diameter distribution of the shaving particles. The simulation results show that in the actual production process, the frequency of the negative pressure fan is at least 27 Hz. Moreover, the paving box should be equipped with a shoulder, and it should be moved back appropriately, but the shoulder distance must be less than 2570 mm. Finally, the particle diameter of the added shavings should be relatively uniform to improve the paving quality of the particleboard.

Future improvements of the model and further analysis should include the following:

1. Improving the model approach and further improving the mechanical structure of the paving box.

2. Validating the model with further experimental data.

## Supporting information

**S1 File.**
(PDF)

## Acknowledgments

The authors are grateful to Nanjing Forestry University for supplying the airflow forming machine and workstation used in this work.

## Author Contributions

**Conceptualization:** Jian Zhang.

**Data curation:** Jian Zhang, Qing Chen.

**Formal analysis:** Jian Zhang, Qing Chen.

**Funding acquisition:** Qing Chen.

**Investigation:** Minghong Shi, Hongping Zhou, Linyun Xu.

**Methodology:** Jian Zhang, Qing Chen, Hongping Zhou.

**Software:** Minghong Shi, Hongping Zhou, Linyun Xu.

**Validation:** Jian Zhang, Qing Chen.

**Writing – original draft:** Jian Zhang, Qing Chen.

**Writing – review & editing:** Jian Zhang, Qing Chen.

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
