## [Decision Letter · Decision Letter 0]

15 May 2021

PONE-D-21-12488

Interaction and Influence of a Flow Field and Particleboard Particles in an Airflow Forming Machine with a Coupled Euler-DPM Model

PLOS ONE

Dear Dr. Chen,

Thank you for submitting your manuscript to PLOS ONE. After careful consideration, we feel that it has merit but does not fully meet PLOS ONE’s publication criteria as it currently stands. Therefore, we invite you to submit a revised version of the manuscript that addresses the points raised during the review process.

We look forward to receiving your revised manuscript.

Kind regards,

Fang-Bao Tian

Academic Editor

PLOS ONE

Journal Requirements:

Reviewers' comments:

Reviewer's Responses to Questions

**Comments to the Author**

1. Is the manuscript technically sound, and do the data support the conclusions?

Reviewer #1: Yes

Reviewer #2: Yes

2. Has the statistical analysis been performed appropriately and rigorously? 

Reviewer #1: Yes

Reviewer #2: Yes

3. Have the authors made all data underlying the findings in their manuscript fully available?

Reviewer #1: Yes

Reviewer #2: Yes

4. Is the manuscript presented in an intelligible fashion and written in standard English?

Reviewer #1: Yes

Reviewer #2: Yes

5. Review Comments to the Author

Reviewer #1: In this paper, the CFD method is employed to investigate the flow field and the gas-solid two-phase flow field coupled with particle movement of an airflow forming machine. The research has practical application value. However there exist some expository short comings as follows:

(1) If you write more details into the Figure 2, it would be better for people to understand this paper. Symbols are marked on the Figure 2, but not in the article. Moreover, the units of length in the figure are missing.

(2) In the text, only cursory information is given on the mesh refinement procedure. Why only the mesh of the air inlet and exhaust outlet is densified? Please give the details of the mesh refinement.

(3) Some details need to be added: the specific location of measuring points 1-7 and the size of the feed opening.

(4) In the fluid-soild coupling simulation, why the particle diameter range of shavings is 0.0002m~0.001m? Can the quality and size of shavings affect the difference between the simulation results and the actual situation? Explain it.

(5) What is the relationship between the airflow field in the third section and the particle field in the fourth section?

(6) How do the added shaving particles affect the speed of the characteristic lines on Sections 2 and 3, please explain it.

(7) English writing needs to be polished. For example, in Page 4 Line 10, ‘The establishment of a CFD simulation model…’ should be corrected. In Page 9 Line 19, ‘The convergence criterion chosen for…’ should be corrected.

Reviewer #2: This manuscript mainly investigates the flow field and the gas-solid two-phase flow field coupled with particle movement of an airflow forming machine by the CFD method. The motivation behind the problem investigated in this manuscript is meaningful and innovative, the manuscript has little short comings as detailed in the follows:

(1) The language needs to be improved sufficiently. For example, in Page 2 Line 6, ‘Particleboards are engineered wood produced by chipping and grinding…’ should be corrected. In Page 14 Line 22, ‘The airflow vortex in the middle of the box almost disappears…’ should be corrected.

(2) In the initial model, why the wind speeds of the air inlet and the exhaust outlet are 10.4 m/s and 24.7 m/s? Explain it.

(3) This paper does not elaborate the choice of calculation model, but directly selects the standardmodel. It is suggested to give reasons.

(4) Improve the analysis of the velocity variation of characteristic lines at different sections.

(5) In Section 4, the quality and shape of particles have a great influence on the results in the DPM simulation. Do you consider the quality and shape of shavings?

6. PLOS authors have the option to publish the peer review history of their article (what does this mean?). If published, this will include your full peer review and any attached files.

Reviewer #1: No

Reviewer #2: No

---

## [Author Response · Author response to Decision Letter 0]

2 Jun 2021

Response to Reviewers

We gratefully thank the editor and all reviewers for their time spend making their constructive remarks and useful suggestions, which has significantly raised the quality of the manuscript and has enable us to improve the manuscript. We have studied comments carefully and have made correction which we hope meet with approval. Below the comments of the reviewers are response point by point and the revisions are indicated.

Reviewer 1

1. Comment: If you write more details into the Figure 2, it would be better for people to understand this paper. Symbols are marked on the Figure 2, but not in the article. Moreover, the units of length in the figure are missing.

1. Reply: Thank you for your comment. We have written more details into the Fig 2 and added the units in the Fig 2 in the revised manuscript. We feel sorry for our carelessness.

2. Comment: In the text, only cursory information is given on the mesh refinement procedure. Why only the mesh of the air inlet and exhaust outlet is densified? Please give the details of the mesh refinement.

2. Reply: As for the referee’s concern, we have added the figure of the mesh refinement in Fig 5. Because the size of the air inlet and exhaust outlet are smaller than that of the whole box, it is necessary to refine the mesh of the two parts to ensure the quality of the mesh, which can speed up the convergence speed of the simulation and improve the accuracy of the simulation.

3. Comment: Some details need to be added: the specific location of measuring points 1-7 and the size of the feed opening.

3. Reply: We are sorry for our negligence of these details. The specific location of measuring points 1-7 is added in Figs 2 and 6. The size of the feed opening is added in Page 6 Line 22 and Fig 3.

4. Comment: In the fluid-solid coupling simulation, why the particle diameter range of shavings is 0.0002m~0.001m? Can the quality and size of shavings affect the difference between the simulation results and the actual situation? Explain it.

4. Reply: Thank you for your question. According to the actual production process, the diameter of shavings used in pavement is about 0.0002 m~0.001m, so the particle diameter range of shavings is 0.0002 m~0.001 m. The quality and size of shavings can’t affect the difference between the simulation results and the actual situation, because the mass flow in the simulation is set to 0.88625 kg/s, which is consistent with the actual production process.

5. Comment: What is the relationship between the airflow field in the third section and the particle field in the fourth section?

5. Reply: Thank you for your question. The simulation analysis of particle field is based on the air flow field. In this study, the particle deposition is simulated to verify that good airflow field characteristics contribute to improving the pavement quality of particleboard, which is mentioned in the fourth section.

6. Comment: How do the added shaving particles affect the speed of the characteristic lines on Sections 2 and 3, please explain it.

6. Reply: We appreciate for your valuable comment. ‘It is obvious from Figs 21 and 22 that Sections 2 and 3 are close to the sudden changed sections of the pavement box, which aggravates the turbulence of Sections 2 and 3. Therefore, the speed of the characteristic lines on Sections 2 and 3 changes obviously due to the addition of shaving particles, but the variation rule of the characteristic line speed in Section 1 is basically unchanged, and the range of velocity variation is stable’ is added in Page 16.

7. Comment: English writing needs to be polished. For example, in Page 4 Line 10, ‘The establishment of a CFD simulation model…’ should be corrected. In Page 9 Line 19, ‘The convergence criterion chosen for…’ should be corrected.

7. Reply: We are very sorry for the mistakes in this manuscript and inconvenience they caused in your reading. ‘The establishment of a CFD simulation model of an airflow forming machine…’ is corrected to ‘The establishment of CFD simulation model of airflow forming machine…’ ‘The convergence criterion chosen for the simulation…’ is corrected to ‘The convergence criteria chosen for the simulation…’ Other errors in English writing have been corrected in the revised manuscript.

Reviewer 2

1. Comment: The language needs to be improved sufficiently. For example, in Page 2 Line 6, ‘Particleboards are engineered wood produced by chipping and grinding…’ should be corrected. In Page 14 Line 22, ‘The airflow vortex in the middle of the box almost disappears…’ should be corrected.

1. Reply: We are very sorry for our incorrect writing in the manuscript. ‘Particleboards are engineered wood produced by chipping and grinding tree logs to obtain wood particles’ is corrected to ‘Particleboards are a kind of engineered wood, produced by chipping and grounding tree’s logs in order to obtain the wood particles’ ‘The airflow vortex in the middle of the box almost disappears…’ is corrected to ‘The air eddies in the middle of the box almost disappear, and the air flows more steadily with the shoulder moved backward’ Other errors in English writing have been corrected in the revised manuscript.

2. Comment: In the initial model, why the wind speeds of the air inlet and the exhaust outlet are 10.4 m/s and 24.7 m/s? Explain it.

2. Reply: Thank you for your question. The initial model is consistent with the actual production model. In the actual production process, the frequencies of the positive pressure fan and negative pressure fan are 20 Hz and 27 Hz, measuring that the wind speeds of the air inlet and the exhaust outlet are 10.4 m/s and 24.7 m/s, respectively.

3. Comment: This paper does not elaborate the choice of calculation model, but directly selects the standard k-ε model. It is suggested to give reasons.

3. Reply: We are appreciative of the reviewer’s suggestion. The air diffusion in the airflow forming machine is slight, and the standard k-ε model is suitable for slight diffusion, which is widely used in industrial flow field simulation, so the standard k-ε model is used to calculate the three-dimensional flow field inside the air paver.

4. Comment: Improve the analysis of the velocity variation of characteristic lines at different sections.

4. Reply: As for the referee’s concern, we have improved the analysis of the velocity variation of characteristic lines at different sections. ‘It is obvious from Figs 21 and 22 that Sections 2 and 3 are close to the sudden changed sections of the pavement box, which aggravates the turbulence of Sections 2 and 3. Therefore, the speed of the characteristic lines on Sections 2 and 3 changes obviously due to the addition of shaving particles, but the variation rule of the characteristic line speed in Section 1 is basically unchanged, and the range of velocity variation is stable’ is added in Page 16.

5. Comment: In Section 4, the quality and shape of particles have a great influence on the results in the DPM simulation. Do you consider the quality and shape of shavings?

5. Reply: Thank you for your question. We have considered the quality and shape of shavings. In the actual production process, the quantity of shavings added is 3190.5 kg/h, so the mass flow is set to 0.88625 kg/s, which has been mentioned in Page 15.

---

## [Editor Report · Decision Letter 1]

3 Jun 2021

Interaction and Influence of a Flow Field and Particleboard Particles in an Airflow Forming Machine with a Coupled Euler-DPM Model

PONE-D-21-12488R1

Dear Dr. Chen,

We’re pleased to inform you that your manuscript has been judged scientifically suitable for publication and will be formally accepted for publication once it meets all outstanding technical requirements.

Kind regards,

Fang-Bao Tian

Academic Editor

PLOS ONE
---

## [Editor Report · Acceptance letter]

7 Jun 2021

PONE-D-21-12488R1 

Interaction and Influence of a Flow Field and Particleboard Particles in an Airflow Forming Machine with a Coupled Euler-DPM Model 

Dear Dr. Chen:

I'm pleased to inform you that your manuscript has been deemed suitable for publication in PLOS ONE. Congratulations! Your manuscript is now with our production department. 

Kind regards, 

on behalf of

Dr. Fang-Bao Tian 

Academic Editor

PLOS ONE